# Contribution of correlated noise and selective decoding to choice probability measurements in extrastriate visual cortex

Yong Gu[1], Dora E Angelaki[2†], Gregory C DeAngelis[3*†]

[1]Institute of Neuroscience and Key Laboratory of Primate Neurobiology, Shanghai Institutes for Biological Sciences, Chinese Academy of Sciences, Shanghai, China; [2]Department of Neuroscience, Baylor College of Medicine, Houston, United States; [3]Department of Brain and Cognitive Sciences, University of Rochester, New York, United States

**Abstract** Trial by trial covariations between neural activity and perceptual decisions (quantified by choice Probability, CP) have been used to probe the contribution of sensory neurons to perceptual decisions. CPs are thought to be determined by both selective decoding of neural activity and by the structure of correlated noise among neurons, but the respective roles of these factors in creating CPs have been controversial. We used biologically-constrained simulations to explore this issue, taking advantage of a peculiar pattern of CPs exhibited by multisensory neurons in area MSTd that represent self-motion. Although models that relied on correlated noise or selective decoding could both account for the peculiar pattern of CPs, predictions of the selective decoding model were substantially more consistent with various features of the neural and behavioral data. While correlated noise is essential to observe CPs, our findings suggest that selective decoding of neuronal signals also plays important roles.

*For correspondence:
gdeangelis@cvs.rochester.edu

†These authors contributed equally to this work

## Introduction

In most sensory systems, neurons encode sensory stimuli by responding selectively to a particular range of stimulus parameters, as typically characterized by tuning curves (*Dayan and Abbott, 2001*). In turn, the pattern of activation across a population of such neurons provides information about the most likely stimulus that may have occurred (*Dayan and Abbott, 2001*). Whether or not a sensory neuron contributes to perceptual decisions generally depends on whether that neuron is selective to the stimulus dimensions relevant to the task at hand, and how much weight is given to the activity of that neuron in population decoding. One method for assessing the potential contribution of a sensory neuron to perception involves measuring the trial-by-trial covariation between neural activity and perceptual decisions, as typically quantified by computing the choice probability (CP) (*Britten et al., 1996*; *Dodd et al., 2001*; *Uka and DeAngelis, 2004*; *Purushothaman and Bradley, 2005*; *Gu et al., 2007, 2008*; *Nienborg and Cumming, 2009, 2010*; *Liu et al., 2013*).

When a sensory neuron shows a significant CP, there is a stereotypical relationship between response, tuning, and choice: neurons tend to respond more strongly when the subject reports perceiving the stimulus as having a value that is more preferred by the neuron. Tested properly, such effects are typically found to be independent of the stimulus value itself (*Britten et al., 1996*; *Uka and DeAngelis, 2004*). While the phenomenology is rather consistent across many studies, the interpretation of CPs has remained controversial (*Nienborg and Cumming, 2010*; *Cohen and Kohn, 2011*). Some studies

**eLife digest** Even the simplest tasks require the brain to process vast amounts of information. To take a step forward, for example, the brain must process information about the orientation of the animal's body and what the animal is seeing, hearing and feeling in order to determine whether any obstacles stand in the way. The brain must integrate all this information to make decisions about how to proceed. And once a decision is made, the brain must send signals via the nervous system to the muscles to physically move the foot forward.

Specialized brain cells called sensory neurons help to process this sensory information. For example, visual neurons process information about what the animal sees, while auditory neurons process information about what it hears. Other sensory neurons—called multisensory neurons—can process information coming from more than one of an animal's senses.

For more than two decades, researchers have known that the firing of an individual sensory neuron can be linked to the decision that an animal makes about the meaning of the sensory information it has received. The ability to predict whether an animal will make a given decision based on the firing of individual sensory neurons is often referred to as a 'choice probability'. Measurements of single neurons have often been used to try to work out how the brain decodes the sensory information that is needed to carry out a specific task. However, it remains unclear whether choice probabilities really reflect how sensory information is decoded in the brain, or whether these measurements are just reflecting coordinated patterns of background 'noise' among the neurons as the decisions are being made.

Gu et al. set out to help resolve this debate by examining choice probabilities in the multisensory neurons in one area of the brain. A series of experiments was conducted to see how these neurons process information, both from the eyes and the part of the inner ear that helps control balance, to work out the direction in which an animal was moving. By performing computer simulations of the activity of groups of neurons, Gu et al. found that choice probability measurements are better explained by the models whereby these measurements did reflect the strategy that is used to decode the sensory information. Models based solely on patterns of correlated noise did not explain the data as well, though Gu et al. suggest that this noise is likely to also contribute to the observed effects.

Following on from the work of Gu et al., a major challenge will be to see if it is possible to infer how the brain extracts the relevant information from the different sensory neurons. This may require recordings from large groups of neurons, but it might help us to decipher how patterns of activity in the brain lead to decisions about the world around us.

---

have suggested that the pattern of CPs across a population of neurons can provide insight into how responses of neurons with different tuning properties are selectively weighted in the decision process, that is selective decoding (*Britten et al., 1996*; *Uka and DeAngelis, 2004*; *Purushothaman and Bradley, 2005*; *Gu et al., 2007*, *2008*). Other studies have pointed out that correlated noise among neurons is necessary to observe significant CPs in large populations, suggesting that CPs are dominated by correlated noise and may not carry any useful information about decoding strategy (*Nienborg and Cumming, 2010*; *Cohen and Kohn, 2011*; *Nienborg et al., 2012*). As an extreme example, neurons that are not involved in the decision process can exhibit significant CPs solely through correlations with other neurons that do contribute (*Cohen and Newsome, 2009*). Thus, a critical issue is whether CPs can reflect selective decoding of sensory neurons.

A recent theoretical study potentially unifies these divergent perspectives (*Haefner et al., 2013*), demonstrating mathematically that CPs could reflect both the structure of correlated noise and selective decoding of neurons. However, experimental evidence that can dissociate these causes has been lacking. Here, we take advantage of a peculiar pattern of CPs exhibited by multisensory neurons that represent translational self-motion (i.e., heading). Some neurons in areas MSTd (*Gu et al., 2006*, *2008*) and VIP (*Chen et al., 2013*) have matched heading preferences in response to visual and vestibular stimuli ('congruent' cells), whereas others prefer widely disparate headings ('opposite' cells). Opposite cells could be decoded such that they provide evidence in favor of either their visual or their vestibular heading preference. We showed previously that congruent and opposite cells have CPs with opposite polarities

in a visual heading discrimination task (*Gu et al., 2008*; *Chen et al., 2013*), and we suggested that this may result from selectively decoding both congruent and opposite cells according to their vestibular heading preferences (*Gu et al., 2008*). This system provides a valuable test bed for exploring the roles of noise correlations and selective decoding in producing CPs.

Using simulations, we explore whether the peculiar pattern of CPs exhibited by multisensory neurons can be explained solely by correlated noise or whether selective decoding is also involved. Our results suggest that selective decoding can play important roles in shaping the pattern of CPs across a population of sensory neurons.

## Results

We explore how selective decoding and correlated noise contribute to choice probabilities (CPs) in three stages. First, we consider a population of hypothetical neurons that represent a stimulus feature, such as heading, based on a single sensory modality (e.g. visual motion). Second, we extend this simplified population model to the multisensory case, in which a second sensory cue (e.g., vestibular) also provides information about the stimulus feature, and we consider the predictions that arise for choice probabilities in neurons with mismatched tuning for the two sensory modalities (opposite cells). Finally, we apply our analyses to models that are based more closely on data from neurons in area MSTd, and we compare the predictions of selective decoding and pure correlation models with the animals' behavior, as well as the structure of correlated noise among neurons.

### Noise correlations and readout for the single modality case

To study how correlated noise and selective decoding affect the CPs of sensory neurons, we first considered a simple model in which only one sensory cue (e.g., visual) was involved. We generated a population of 1000 hypothetical neurons with cosine tuning for heading. All neurons in this population had tuning curves with the same amplitude and width, but differed in their heading preferences. For simplicity, half of the neurons preferred leftward heading (−90°) while the other half preferred rightward heading (+90°) so that tuning curves for any pair of neurons had either identical slopes or opposite slopes around a straight-forward heading reference. Other distributions of heading preferences (e.g., uniform or bimodal) did not substantially alter our conclusions. We used a maximum likelihood decoder (*Sanger, 1996*; *Dayan and Abbott, 2001*; *Jazayeri and Movshon, 2006*; *Gu et al., 2010*; *Fetsch et al., 2011*) to estimate heading from simulated population activity, and we required the population activity to discriminate between headings that were slightly leftward or rightward relative to a straight-forward reference heading. Specifically, a likelihood function over heading was computed from the population activity on each trial. The decoder then made a 'leftward' choice if the area under the likelihood function for leftward headings exceeded the area under the curve for rightward headings, and vice versa for a 'rightward' choice. We then computed each model neuron's CP for the ambiguous stimulus condition (i.e., straight forward motion, 0°, 'Materials and methods'), and we explored how correlated noise and selective decoding affected CPs.

As a prelude to considering the multisensory situation, we consider two extreme cases in which CPs of a group of neurons are driven mainly by correlated noise or by selective decoding. In both schemes, structured noise correlations are necessary to observe significant CPs, but the models differ in terms of which pools of neurons are correlated and how they are decoded. For the 'pure-correlation' model (*Figure 1A,B*), only correlated noise is needed to produce CPs that are significantly different from the chance level of 0.5. In this model, we divided the population of neurons into two groups, each of which contained an equal number of neurons preferring leftward and rightward headings. The first group of neurons (pool 1 in *Figure 1A*) contributed to the decoder's heading report (decoding weight = 1), while responses from the other group (pool 2) were ignored by the decoder (decoding weight = 0). We then examined the CPs of pool 2 neurons as a function of their correlations with pool 1. Although the signals from pool 2 neurons did not contribute to the decoder output, they still exhibited significant CPs as long as their noise was correlated with that of pool 1 neurons (*Cohen and Newsome, 2009*).

We introduced correlated noise having a structure that is based on experimental observations from heading-selective neurons in both cortical (*Gu et al., 2011*; *Chen et al., 2013*) and subcortical (*Liu et al., 2013*) areas. Specifically we assumed that noise correlations among pairs of model neurons are a linear function of their tuning similarity, or signal correlation ($r_{signal}$, *Figure 1B*, insets). In this correlation structure, neurons with similar heading tuning (positive $r_{signal}$) generally have positive noise correlations, whereas neurons with dissimilar tuning (negative $r_{signal}$) tend to have negative noise correlations. This

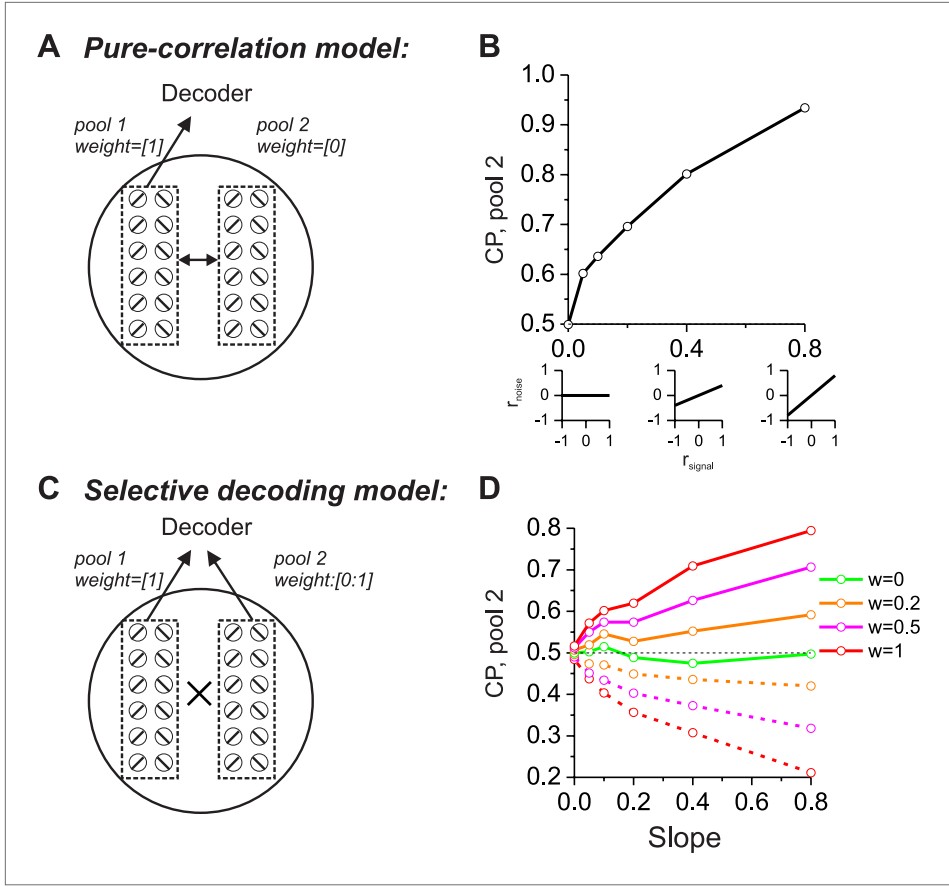

**Figure 1**. Comparison of models in which choice probabilities (CPs) arise through either correlated noise or selective decoding. Each model consists of two pools of neurons (500 neurons each) with equal numbers of neurons that prefer leftward and rightward headings. In the 'pure-correlation' model (**A** and **B**), neurons in pool 2 make no contribution to the decision and activity within or across pools is correlated according to the relationship illustrated in panel **B**. In the 'selective decoding' model (**C** and **D**), neurons shared correlated noise within each pool but not across pools. Neurons in pool 1 were always given a decoding weight of 1, while neurons in pool 2 were given weights ranging from 0 to 1. Solid curves in **D**: responses of pool 2 were decoded according to each neuron's preferred stimulus; dashed curves: pool 2 responses were decoded relative to each neuron's anti-preferred stimulus. Dashed black horizontal line: CP = 0.5.

correlation structure was applied to all pairs of neurons, regardless of whether they were from pool 1 or pool 2. For the simulations, the average noise correlation across all neurons was close to zero, consistent with some previous findings (*Gu et al., 2011*), but the results are not sensitive to this mean value (*Liu et al., 2013*). Rather, the critical factor is the slope of the relationship between $r_{noise}$ and $r_{signal}$ (*Liu et al., 2013*): larger slopes lead to greater CPs among pool 2 neurons (*Figure 1B*). Hence, in this pure-correlation model, CPs of pool 2 neurons are driven exclusively through correlations with neurons in pool 1 that contribute to the decision process (*Cohen and Newsome, 2009*; *Nienborg and Cumming, 2010*).

For the 'selective decoding' model, noise among neurons within each pool was correlated in the same manner as described above, but there were no correlations between neurons in different pools (*Figure 1C*). In this case, significant CPs for pool 2 neurons require that these neurons make a contribution to the decision (*Figure 1D*). We manipulated two aspects of the contribution of each pool 2 neuron to the decoder output: magnitude and polarity. The magnitude reflects how strongly each neuron's activity influenced the decoder and was implemented mathematically by multiplying each neuron's contribution by a value in the range [0–1]. If the weight value is 1, then pool 2 neurons contribute equally as pool 1 neurons. As the weight is reduced toward zero, the contribution of pool 2 neurons diminishes and eventually is eliminated. The polarity (or sign) of the weight determines

whether each neuron provides the decoder with evidence for (positive polarity) or against (negative polarity) its heading preference. It may appear counterintuitive to consider that neurons might provide evidence against their stimulus preference, but the need to consider this case will arise later in the multisensory version of the model due to the presence of opposite cells. These neurons have different heading preferences for the two sensory modalities, so they can provide evidence in favor of their stimulus preference for one modality or the other.

In this selective decoding model, the magnitude of CP increases with the weight applied to pool 2 neurons. In addition, whether the CP value is greater or less than 0.5 depends on the polarity of the contribution of pool 2 neurons (*Figure 1D*). Interpreting responses as evidence in favor of the preferred heading produces CP >0.5 (solid curves) while decoding responses as evidence against the preferred heading leads to CP <0.5 (dashed curves). Hence, the two models generate CPs for pool 2 neurons through different mechanisms. In the pure-correlation model, CPs of pool 2 neurons are produced through correlations with pool 1 neurons that are involved in the decision process (*Figure 1B*). In the selective decoding model, pool 2 neurons have CPs that depend on how strongly they contribute to the decision, as well as the polarity of the contribution of each neuron to the decision (*Figure 1D*).

## Noise correlations and readout for the multiple modality case

We next consider a more complicated case in which two different sensory cues are involved in a perceptual decision. For example, both visual (optic flow) and vestibular signals provide information about the direction of self-motion, or heading (*Angelaki and Cullen, 2008*; *Britten, 2008*). Previous studies have reported that neurons in multiple cortical areas (e.g., MSTd, VIP, VPS) are tuned for heading, and tend to prefer either the same or opposite headings defined by optic flow and vestibular cues (*Page and Duffy, 2003*; *Gu et al., 2006*, *2008*; *Chen et al., 2011a*, *2011b*). We refer to these as congruent cells and opposite cells, respectively (*Figure 2A*). For opposite cells, the preferred heading is different for the two sensory modalities, thus raising the fundamental question of how these cells may be decoded. In a multimodal heading discrimination task (*Gu et al., 2008*), we showed previously that CPs of MSTd neurons have a peculiar dependence on the congruency of visual/vestibular heading tuning (*Figure 2B*). For congruent cells (cyan symbols in *Figure 2B*), CPs were consistently >0.5 when heading judgments were based on either vestibular or visual cues. In contrast, CPs for opposite cells tended to be >0.5 in the vestibular task condition but <0.5 in the visual condition (magenta symbols in *Figure 2B*). We suggested previously (*Gu et al., 2008*) that this peculiar pattern of CPs might result from decoding the responses of MSTd neurons according to their vestibular heading preferences, a form of selective decoding. In what follows, we evaluate whether this pattern of CPs is compatible with a multisensory version of either the pure correlation or selective decoding models described above.

We again divided 1000 model neurons into two pools of equal size, with one pool consisting of congruent cells and the other consisting of opposite cells (*Figure 2C,E*). For simplicity, these neurons again preferred either leftward or rightward headings. In the pure-correlation model, only pool 1 neurons (congruent cells) provided inputs to the decoder's decision process, whereas pool 2 neurons (opposite cells) were given no weight. However, opposite cells can still exhibit CPs as long as they are correlated with congruent cells (*Figure 2D*). For a pair of neurons that includes an opposite cell, correlated noise could depend on the similarity of vestibular tuning between the members of the pair or on the similarity of visual tuning (or on both, as discussed further below). If correlated noise is dependent on the similarity of vestibular tuning (i.e., vestibular signal correlation), opposite cells show CPs that are >0.5 in the vestibular condition and <0.5 in the visual condition (*Figure 2D*), roughly similar to experimental data from MSTd neurons (*Figure 2A,B*). The intuition for this result is straightforward. When a congruent cell fires more spikes than average, an opposite cell that shares the same vestibular tuning will also fire more spikes than average and will have a CP >0.5 in the vestibular condition. In the visual condition, this correlation structure will lead to an opposite cell responding more when the animal chooses its non-preferred visual heading, thus producing a CP <0.5. Note that this prediction of the pure correlation model only mimics real data if opposite cells are correlated with congruent cells having matched vestibular heading preferences.

In contrast, if correlated noise depends on the similarity of visual heading preferences, then the pattern of results will be reversed for the vestibular and visual conditions (*Figure 2—figure supplement 1*), which is clearly inconsistent with the experimental data (*Figure 2B*). Finally, correlations between the two pools of the pure-correlation model could depend on the similarity of heading

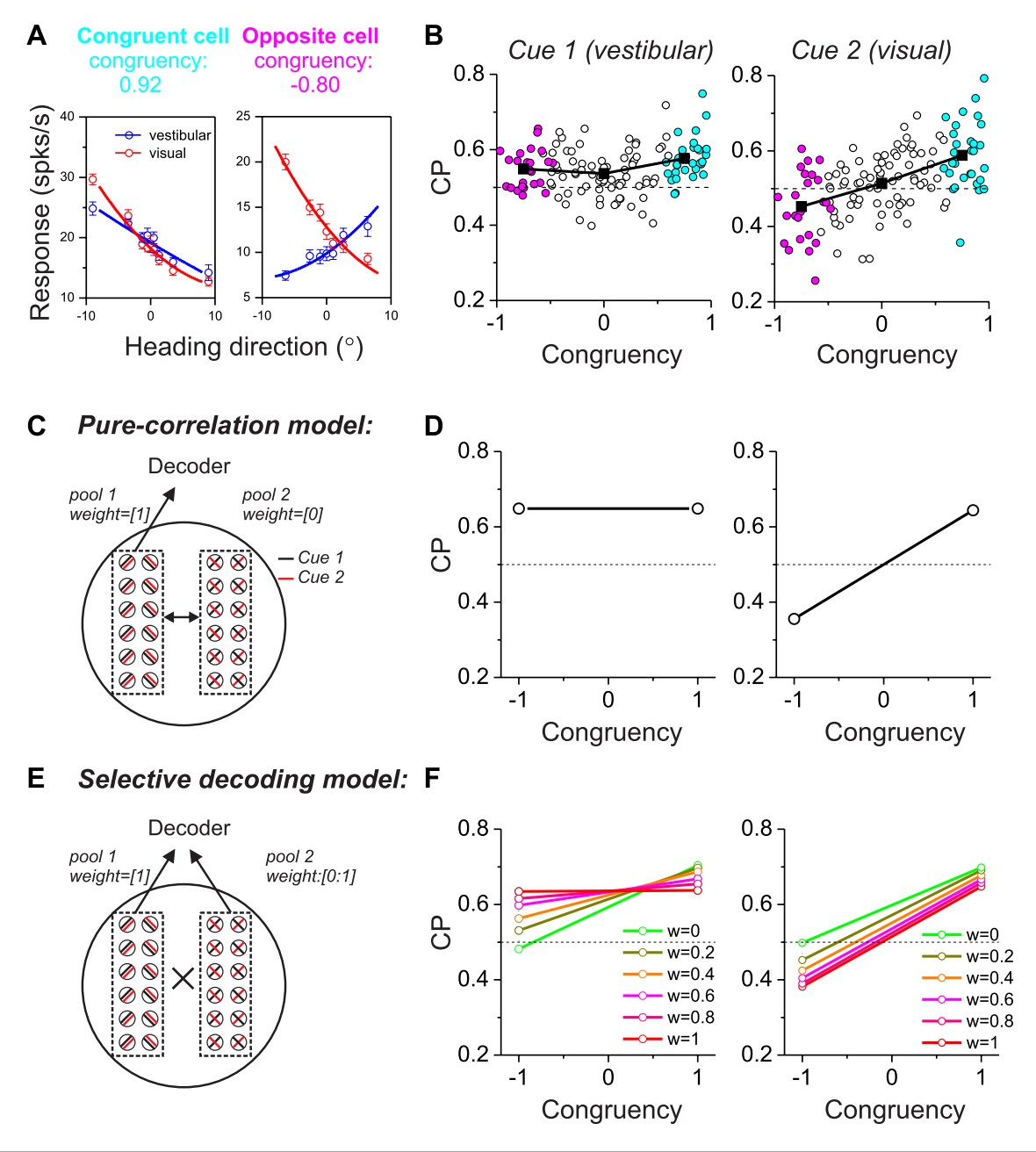

**Figure 2**. Responses of multisensory neurons and multisensory versions of the pure-correlation and selective-decoding models. (**A**) Heading tuning curves from two example MSTd neurons measured during a fine heading discrimination task (*Gu et al., 2008*): one congruent cell (left) and one opposite cell (right). Red and blue data show responses measured for the visual and vestibular conditions, respectively. (**B**) Choice probability as a function of congruency for MSTd neurons tested in the vestibular (left) and visual (right) conditions (adapted, with permission, from Supplement Figure 8A and Figure 6C of *Gu et al.(2008)*; respectively). Cyan and magenta symbols denote data for congruent and opposite cells, respectively (unfilled symbols: intermediate cells). (**C** and **D**) Multisensory version of the pure-correlation model (500 model neurons in each pool). Pool 1 consists of all congruent cells (same slope tuning curves for the two cues), whereas pool 2 contains all opposite neurons. Correlated noise within or across pools depends only on the similarity of tuning for cue 1. (**E** and **F**) In the selective decoding model, neurons were correlated according to the similarity of tuning for both cues ('Materials and methods'). This rule generated correlated noise within each pool but not between pools. Neurons in pool 1 were always given a full weight of 1 in the decoding, whereas the decoding weights of neurons in pool 2 ranged from 0 to 1 (different colors in **F**). Dashed black horizontal line: CP = 0.5.

The following figure supplements are available for figure 2:

*Figure 2. Continued on next page*

*Figure 2. Continued*

**Figure supplement 1**. Predictions from a variant of the pure-correlation model in which correlated noise depends only on signal correlations from the visual tuning curves.

**Figure supplement 2**. Predictions from a variant of the selective decoding model in which responses are decoded according to the visual heading tuning of each neuron, instead of the vestibular tuning.

**Figure supplement 3**. Predictions from a "hybrid" model (see text for details) in which correlated noise was assigned according to vestibular signal correlations, and heading was decoded relative to the vestibular heading tuning of each neuron.

preferences for both the visual and vestibular modalities ('Materials and methods'; *Equation 2*). If correlated noise depends equally on both visual and vestibular signal correlations (i.e., *Equation 2* with $a_{vestibular} = a_{visual}$), then correlations between opposite cells and congruent cells become effectively zero because the two terms of *Equation 2* cancel. In this case, the pure-correlation model becomes equivalent to the selective decoding model with zero weight placed on opposite cells, as considered below.

In the selective decoding model, we assume that correlated noise among a pair of neurons depends on the similarity of both vestibular and visual tuning ('Materials and methods'), as demonstrated previously for pairs of MSTd neurons (*Gu et al., 2011*). Because we assume $a_{vestibular} = a_{visual}$ in *Equation 2*, the resulting effective noise correlation between mixed pairs of congruent and opposite cells will be zero (denoted by ' × ' between the two pools in *Figure 2E*). Thus, use of the selective-decoding model with this particular correlation structure effectively eliminates correlated noise between the two pools while allowing the decoding weights of pool 2 to vary. Under these conditions, non-zero decoding weights must be applied to pool 2 in order to observe CPs for opposite cells (*Figure 2F*). Note that, for the selective decoding model, we do not consider situations in which correlated noise between the two pools depends only on vestibular or visual heading preferences. In such cases, there would be correlated noise between pairs of neurons drawn from the two pools, and the CPs of opposite cells would depend on both readout weights and noise correlations. Thus, the two models would no longer be conceptually distinct. We shall evaluate these assumptions in the following sections.

To examine the predictions of the selective decoding model, we again manipulated two aspects of the readout. With regard to magnitude of the decoding weights, a decoding weight different from zero was essential to produce CPs for opposite cells that were different from the chance level (green symbols/lines vs other colors in *Figure 2F*). With regard to polarity (or sign) of the decoding weights, decoding responses of opposite cells with respect to their vestibular heading preference led to CP >0.5 in the vestibular condition and <0.5 in the visual condition (*Figure 2F*), which was similar to that seen in the real data. On the other hand, if responses of model neurons were decoded with respect to their visual heading preferences, the pattern of CPs across stimulus conditions would reverse and would be incompatible with that observed for MSTd neurons (*Figure 2—figure supplement 2*).

Hence, both models can produce CP patterns for congruent and opposite cells that are similar to those seen in the real data, but through critically different mechanisms. In the pure-correlation model, CPs of opposite cells arise solely through correlations with congruent cells having matched vestibular tuning, even though opposite cells do not contribute directly to the decision. In the selective decoding model, there is no effective correlation between mixed pairs of congruent/opposite cells (i.e., no correlated noise between pools). In this case, to qualitatively match the pattern of CP results from real neurons, the activity of opposite cells must be given weight in the decoding and these cells must be decoded selectively according to their vestibular preferences. Another way to summarize the distinction between models is that the pure correlation model mimics experimental CP results by virtue of modality specificity in the structure of correlated noise, whereas the selective decoding model achieves this by modality specificity in the decoding weights.

For completeness, we also considered a hybrid model that combines features from both of the above models. Noise correlations were linearly dependent only on the similarity of vestibular tuning, as in the pure-correlation model. This produced correlations between the two pools (congruent and opposite cells), unlike in the selective decoding model. In addition, a readout weight was assigned to the opposite cells, as in the selective decoding model. Under these conditions, we found that the predicted patterns of CPs largely resembled those from the pure-correlation model, as if the decoding

weights on pool 2 played little role (*Figure 2—figure supplement 3*). In the following analyses, we only considered comparisons between the pure-correlation model and the selective decoding model, as these are conceptually distinct. Although this distinction is useful for exploring the relative roles of correlated noise and selective decoding in producing CPs, we recognize that both factors may contribute.

## Comparison of model predictions with data: noise correlations

Which model best matches experimental data on correlated noise? Our previous study (*Gu et al., 2011*) showed that $r_{noise}$ measured for pairs of MSTd neurons depended approximately equally on $r_{signal}$ computed from both visual and vestibular tuning curves (*Figure 3A*). This dependence on $r_{signal}$ for both modalities is not due to strong covariance between the two signal correlations because $r_{signal}$ values for visual and vestibular tuning are only weakly correlated (*Figure 3—figure supplement 1*). Note also that noise correlations are generally negative for neurons with opposite tuning (negative signal correlations) in our heading discrimination data sets (*Gu et al., 2011*; *Chen et al., 2013*; *Liu et al., 2013*), such that subtracting responses of oppositely tuned neurons is not expected to have the benefit seen in other systems (*Romo et al., 2003*).

Here, we have sorted these previous data (*Gu et al., 2011*) into two groups according to whether the congruency of visual/vestibular tuning is matched or mismatched for the two members of each pair of neurons: 'matched congruency' pairs consist of either two congruent cells or two opposite cells, whereas 'mismatched congruency' pairs consist of one congruent cell and one opposite cell (see 'Materials and methods' for classification procedure). Pairs that could not be classified into either group were labeled as 'undefined' (open symbols, *Figure 3A*). We found that the dependence of $r_{noise}$ on both visual and vestibular signal correlations was weak for the mismatched congruency pairs (vestibular $r_{signal}$: slope = 0.033, CI = [−0.5−0.27], R = 0.08, p=0.8; visual $r_{signal}$: slope = −0.057, CI = [−0.424−0.258], R = −0.12, p=0.7, type II linear regression), whereas this dependence was quite robust for the matched congruency pairs (vestibular $r_{signal}$: slope = 0.192, CI = [0.082−0.288], R = 0.61, p=0.001; visual $r_{signal}$: slope = 0.186, CI = [0.069−0.28], R = 0.62, p<0.001, type II linear regression). This pattern of results is consistent with the correlation structure assumed in the selective decoding model.

To evaluate whether the experimental data significantly favor the selective decoding model over the pure correlation model, we fit the data from these 127 pairs of neurons with two correlation structures: (1) $r_{noise}$ depended only on vestibular $r_{signal}$, $r_{noise} = a_{vestibular} * r_{signal\_vestibular}$, as in the pure correlation model; (2) $r_{noise}$ depended on both vestibular and visual signal correlations, $r_{noise} = a_{vestibular} * r_{signal\_vestibular} + a_{visual} * r_{signal\_visual}$, as assumed in the selective decoding model. We then used linear regression to fit the data with both correlation structures. Importantly, we found that the model with coefficients for both vestibular and visual signal correlations provided a significantly better fit to the data, after accounting for the difference in the number of parameters (p=0.0003, sequential F-test). The coefficients of the best-fitting model were $a_{vestibular} = 0.12$ and $b_{visual} = 0.09$, respectively. Thus, the empirical correlation data are significantly better fit with a correlation structure in which $r_{noise}$ depends on $r_{signal}$ for both vestibular and visual tuning curves.

To help visualize why the second structure above better fits the data, we used the measured signal correlations for each pair of neurons, along with the fitted coefficients, to predict $r_{noise}$ for each pair of neurons. When $r_{noise}$ is predicted based on both visual and vestibular signal correlations (*Figure 3B*), the dependence of $r_{noise}$ on $r_{signal}$ was much weaker for mismatched congruency pairs (vestibular $r_{signal}$: slope = 0.074, 95% CI = [0.044−0.109], R = 0.82, p=0.004; visual $r_{signal}$: slope = −0.01, 95% CI = [−0.08−0.066], R = −0.09, p=0.8, type II linear regression) than for matched congruency pairs (vestibular $r_{signal}$: slope = 0.19, 95% CI = [0.177−0.203], R = 0.99, p<<0.001; visual $r_{signal}$: slope = 0.18, 95% CI = [0.157−0.195], R = 0.96, p<<0.001, type II linear regression), similar to the MSTd data (*Figure 3A*). In contrast, when $r_{noise}$ is only dependent on vestibular $r_{signal}$, the predicted correlation structure is quite different (*Figure 3C*). While $r_{noise}$ is perfectly correlated with vestibular $r_{signal}$ for all neurons (by assumption), the mismatched congruency pairs reveal roughly equal but opposite dependencies on vestibular and visual signal correlations (vestibular $r_{signal}$: slope = 0.16, 95% CI = [0.161−0.161], R = 1, p<<0.001; visual $r_{signal}$: slope = −0.12, 95% CI = [−0.220−0.018], R = −0.65, p=0.04, type II linear regression). Thus, the available data on noise and signal correlations compare more favorably with the assumptions of the selective decoding model than the pure correlation model.

## Comparison of model predictions with data: choice probabilities

We now evaluate whether predictions of the pure correlation and selective decoding models are compatible with choice probability data obtained from area MSTd neurons during a fine heading

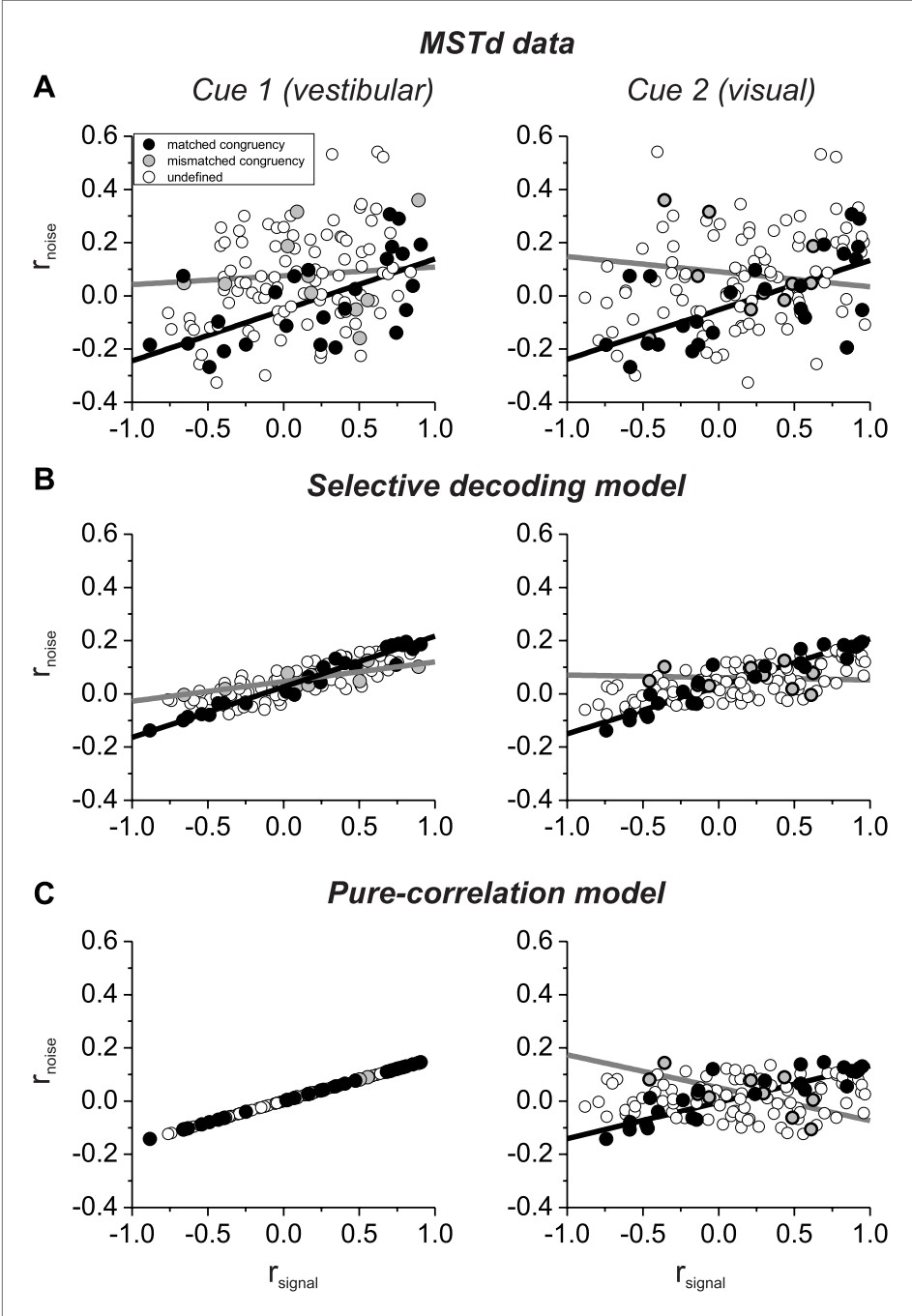

**Figure 3**. Comparison of the structure of correlated noise between models and data. (**A**) Data from pairs of neurons recorded from area MSTd (*Gu et al., 2011*). Noise correlation is plotted against signal correlations obtained from the vestibular (left column) or visual (right column) tuning curves. Black and gray symbols denote pairs with matched (black) or mismatched (gray) congruency. Open symbols represent undefined pairs. (**B**) Predicted noise correlations as a function of signal correlation based on fits of the selective decoding model. Format as in panel **A**. (**C**) Predicted noise correlations as a function of signal correlation by for the pure-correlation model fit, for which noise correlations depend only on vestibular signal correlation.

The following figure supplements are available for figure 3:

**Figure supplement 1**. Comparison of vestibular and visual signal correlations for 127 pairs of neurons simultaneously recorded from area MSTd by *Gu et al. (2011)*.

*Figure 3. Continued*

**Figure supplement 2**. Noise correlation structure of the pure correlation model computed from the signal correlations of all distinct pairings of 129 neurons that were recorded previously by *Gu et al. (2011)*.

**Figure supplement 3**. Noise correlation structure of the selective decoding model computed from the signal correlations of all distinct pairings of 129 neurons that were recorded previously by *Gu et al. (2011)*.

discrimination task (*Gu et al., 2008*). The Gu et al. dataset consisted of 129 single neurons that were not recorded simultaneously. Thus, to generate model population responses for decoding analyses, we generated responses for a model population of 1000 neurons, each of which had visual and vestibular tuning curves that were obtained by drawing data (with replacement) from the sample of 129 real MSTd neurons. Responses of the 1000 model neurons on each simulated trial were generated from a covariance matrix that was based on the two different correlation structures described in the previous section, with parameters that were obtained by fits to the MSTd data. This yielded noise and signal correlations similar to those described in *Figure 3* (*Figure 3—figure supplements 2 and 3*). With these constraints, we can decode the simulated population responses ('Materials and methods') and make predictions of CPs and neuronal thresholds.

With our biologically-constrained versions of the pure correlation and selective decoding models, we now consider the patterns of choice probabilities predicted by each model and how they compare to data from MSTd neurons. In the pure-correlation model (*Figure 4A*), the average CP for congruent cells is significantly greater than 0.5 for both the vestibular (0.65 ± 0.06 SD) and visual (0.65 ± 0.04 SD) conditions (p<0.001, *t* test). For opposite cells, the average CP is significantly >0.5 in the vestibular condition

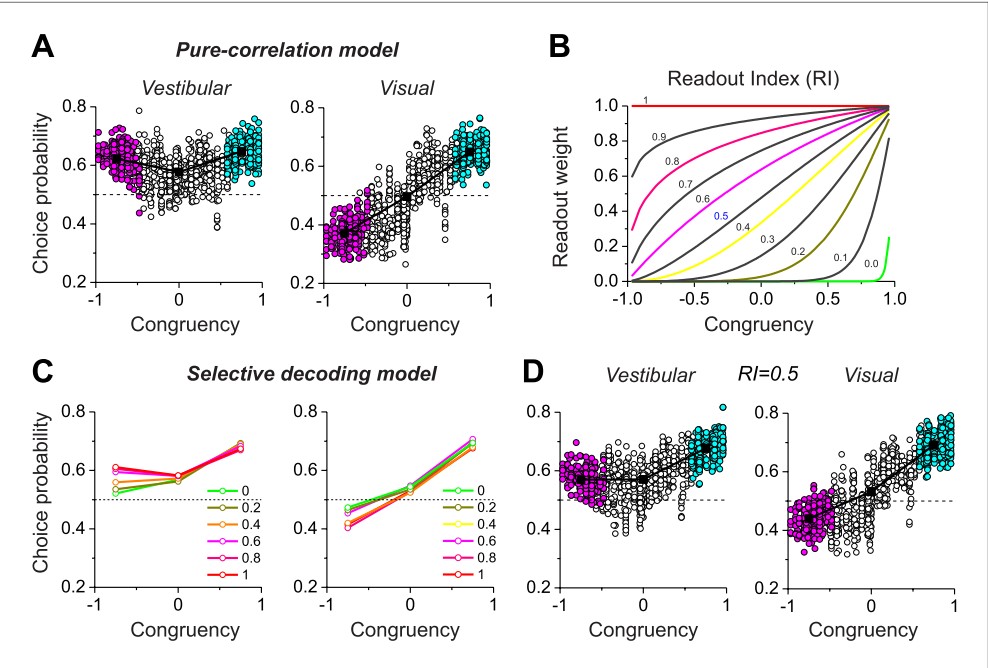

**Figure 4**. Predictions of choice probabilities from the two models. (**A**) The pattern of CPs predicted by the pure-correlation model. Format as in *Figure 2B*. (**B**) A family of weighting profiles used to consider various degrees of contribution of opposite cells to the selective decoding model. Each curve shows the decoding weight as a function of congruency between visual and vestibular heading tuning. Each curve corresponds to a specific value of the Readout Index (RI). (**C**) Predicted average CPs from the selective decoding model for a subset of the RI values illustrated in (**B**). (**D**) The pattern of CPs across neurons in the selective decoding model for an RI value of 0.5. Cyan symbols: congruent cells; Magenta symbols: opposite cells; Unfilled symbols: intermediate cells. Solid squares: mean CP. Dashed horizontal line: CP = 0.5.

(0.623 ± 0.039 SD, p<0.001, $t$ test) and significantly <0.5 in the visual condition (0.372 ± 0.047 SD, p<0.001). This pattern of CPs is qualitatively similar to that observed for MSTd neurons (*Figure 2B*).

In the selective decoding model, we need to consider how the weight applied to opposite cells influences the pattern of CPs. However, unlike the simple hypothetical model of *Figure 2*, real neurons exhibit a range of congruencies between visual and vestibular tuning. To explore a range of possible relative weightings of opposite cells, we used a sigmoidal function (having a single parameter called the Readout Index, RI) to selectively weight the contributions of congruent and opposite cells (*Figure 4B*, 'Materials and methods'). For RI values near 1.0, all neurons contribute equally to the decoder output, regardless of congruency. As RI declines, opposite cells are given gradually less weight in the decoding (*Figure 4B*). For small values of RI, opposite neurons in the selective decoding model show CPs that approach 0.5, as expected since these neurons are given little weight in the decision process. However, as RI increases, the average CP for opposite cells increases in the vestibular condition and decreases in the visual condition (*Figure 4C*). For RI values near 0.5 (*Figure 4D*), the pattern of CPs across the population of simulated neurons resembles that exhibited by MSTd neurons (*Figure 2B*).

Together, these results (*Figure 4*) demonstrate that both the pure correlation and selective decoding models are capable of mimicking the patterns of CPs shown by MSTd neurons in the visual and vestibular conditions (although the pure correlation model assumes a correlation structure different from that seen experimentally). We now consider whether the models make distinct predictions regarding CPs for the combined condition, in which visual and vestibular cues are presented together. As described previously (*Gu et al., 2008*), when animals judge heading based on congruent combinations of visual and vestibular cues, congruent cells tend to have CPs >0.5, whereas opposite cells have CPs that cluster around the chance level of 0.5 (*Figure 5A*). Interestingly, although the average CP of opposite cells was approximately 0.5 for both models in the combined condition (pure-correlation model: 0.486 ± 0.13 SD; selective decoding model: 0.491 ± 0.06 SD), the SD of CP across the population of opposite cells was much greater for the pure-correlation model than for the selective decoding model (*Figure 5B,C*). Indeed, a non-parametric test showed that the dispersion of the CPs was significantly greater for the pure correlation model (p<<0.001, Ansari–Bradley test). The reason for this difference is apparent: CPs of opposite cells in the pure correlation model clearly have a bimodal distribution ($p_{uni}$<<0.001, $p_{bi}$>0.05, modality test, middle column of *Figure 5D*). In contrast, CPs of opposite cells in the selective readout model have a unimodal distribution centered around 0.5 ($p_{uni}$>0.05, modality test, right column in *Figure 5D*), which is similar to that seen for MSTd neurons ($p_{uni}$>0.05, modality test, left column in *Figure 5D*). Thus, the pattern of CPs in the combined condition favors the selective decoding model.

Additional analyses suggest that this difference in the shape of the CP distribution between models is fairly robust to variations in the key model parameters. For the pure correlation model, a bimodal distribution of CPs in the combined condition is a robust result over a range of values of $a_{vestibular}$, including values much smaller than needed to fit our noise correlation data (*Figure 5—figure supplement 1*). For the selective decoding model, a unimodal distribution of CPs is predicted for a wide range of RI values, and significant bimodality only occurs for RI values ≥0.8, which are not consistent with behavioral thresholds as described below (*Figure 5—figure supplement 2*).

A possible explanation for the near-chance CPs of opposite cells in the combined condition is that the sensitivity these cells is low relative to that in the single cue conditions and thus opposite cells contribute less to the decision process (see insets in *Figure 5A*, left panel). Indeed, for real MSTd neurons, congruent cells tend to have low thresholds and high CPs whereas opposite cells tend to have high thresholds and low CPs (*Figure 5E*; R = −0.65, 95% CI: [−0.47 to −0.78], p<<0.001, linear regression). Although both models also show a similar dependency of CP on neuronal sensitivity in the combined condition (p<<0.001, linear regression, *Figure 5F,G*), the strength of the correlation is significantly weaker for the pure correlation model (R = −0.41, 95% CI: [−0.32 to −0.48]) than for the selective readout model (R = −0.67, 95% CI: [−0.61 to −0.72], *Figure 5H*). This also appears to be largely due to the bimodal distribution of CPs for opposite cells in the pure correlation model, which is not observed for the real MSTd data. Thus, while both models produce similar patterns of CPs in the vestibular and visual conditions (*Figure 4*), the pattern of CPs from the selective decoding model was more analogous to the measured data in the combined condition.

## Comparison of task performance between models and monkeys

Finally, we compute the predicted psychophysical performance by decoding data from real MSTd neurons using each readout model, and we compare the results to animals' behavioral performance. In a fine heading discrimination task, we have previously shown that heading sensitivity in the combined

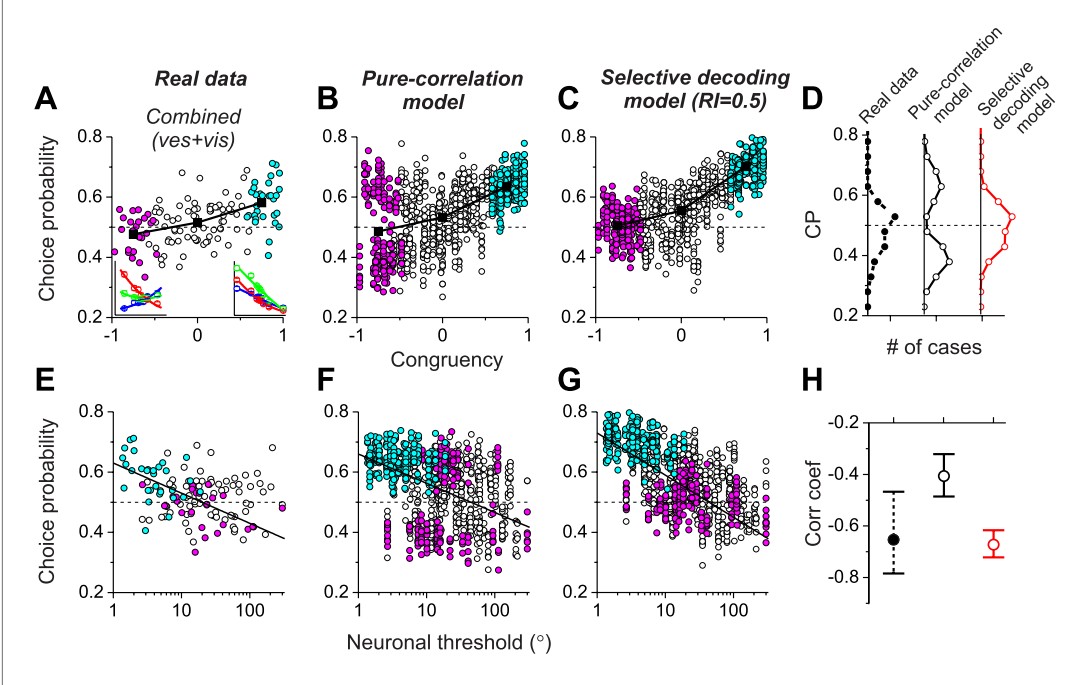

**Figure 5**. Analysis of choice probabilities for the combined condition in which both visual and vestibular heading cues are present. (**A**) CP values plotted as a function of congruency index for neurons from area MSTd tested in the Combined condition (adapted, with permission, from *Figure 6A* of *Gu et al., 2008*). (**B** and **C**) CP as a function of congruency for model neurons from the pure-correlation model, and model cells from the selective decoding model, respectively. (**D**) Distributions of CPs for opposite cells from area MSTd, the pure-correlation model and the selective decoding model. (**E**–**G**) CP values plotted as a function of neuronal discrimination thresholds for real MSTd neurons (adapted, with permission, from *Figure 6B* of *Gu et al., 2008*), units from the pure-correlation model and units from the selective decoding model, respectively. (**H**) Correlation coefficient of the best linear fit to the relationship between CP and neuronal threshold for MSTd data (filled black circles), pure-correlation model (open black circles) and selective decoding model (open red circles). Error bars represent 95% confidence intervals.

The following figure supplements are available for figure 5:

**Figure supplement 1**. Bimodality of CP for opposite cells in the cue combined condition.

**Figure supplement 2**. Same format as in *Figure 5—figure supplement 1*, but results are shown for the selective decoding model.

condition increases when the two cues have comparable sensitivity, and that the effects are close to that predicted from optimal cue integration theory (*Gu et al., 2008*; *Fetsch et al., 2009*, *2011*). In *Figure 6A*, we have replotted the animals' behavioral data (solid circles and dashed curve). The average psychophysical thresholds were 2.03°±0.09° (mean ± SEM) and 2.12° ± 0.1° for the vestibular and visual conditions, respectively. In the combined condition, the average threshold was reduced to 1.44° ± 0.06°, a 29% improvement compared to the best single cue, and was very close to the prediction (1.43° ± 0.06°) from optimal cue integration theory.

Unlike the animals' behavior, decoder performance based on the pure-correlation model predicted mismatched thresholds for the visual and vestibular conditions (*Figure 6A*, open symbols and solid black curve). The average threshold predicted for the visual condition (1.24° ± 0.02°, mean ± SEM) was 42% lower than that for the vestibular condition (2.16° ± 0.14°, mean ± SEM). This may be due to the fact that the average neuronal threshold of congruent MSTd neurons tends to be lower for the visual condition (5.5°) than the vestibular condition (7.1°), although this difference did not reach significance (p=0.106, *t* test, N = 30, *Figure 6—figure supplement 1*). Despite the performance mismatch between the two single-cue conditions, sensitivity was still improved during the combined condition (1.10° ± 0.05°), as expected by optimal cue integration predictions (1.07° ± 0.02°).

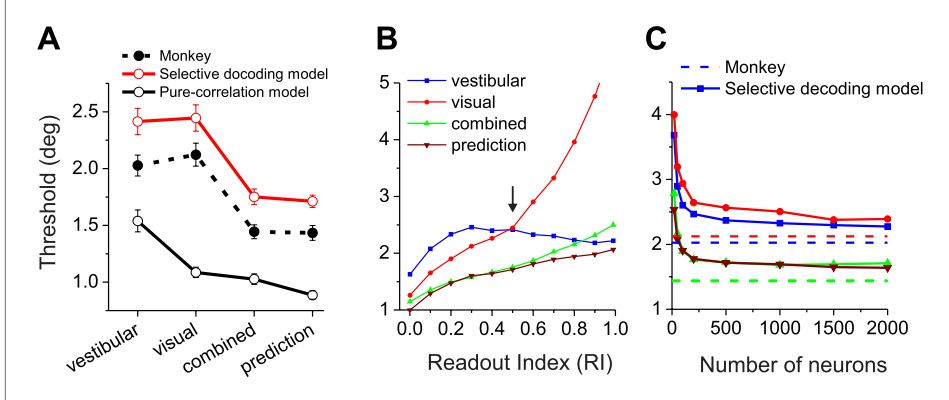

**Figure 6**. Comparisons between model population threshold and psychophysical performance of the animals. (**A**) Average thresholds are shown for the vestibular, visual, and combined conditions, along with the prediction from optimal cue integration theory. Data are shown for the average of two monkeys (filled symbols, dashed curve), for predictions of the pure-correlation model (black open symbols and solid curve), and for predictions of the selective decoding model with RI = 0.5 (red symbols and curve). (**B**) Predicted thresholds from the selective decoding model are plotted as a function of Readout Index for vestibular, visual, and combined conditions, as well as the optimal prediction from the single-cue thresholds. (**C**) Thresholds from the selective-decoding model (with RI = 0.5) as a function of population size. Solid curves: model predictions; dashed horizontal lines: average performance of two animals.

The following figure supplements are available for figure 6:

**Figure supplement 1**. Comparison of neuronal sensitivity between the visual (ordinate) and vestibular (abscissa) stimulus conditions for 30 congruent cells.

For the selective decoding model, we examined how performance changes as a function of the weighting applied to opposite cells (Readout Index, *Figure 6B*, 'Materials and methods'). In the vestibular condition, the decoder's discrimination threshold decreased modestly as more weight was applied to opposite cells because more cells contribute to the likelihood function (*Figure 6B*, blue curve). In the visual condition, because heading was decoded with respect to the vestibular preference of each neuron, a greater contribution of opposite cells tends to drive the decoder to make choices that are opposite to that being signaled by congruent cells. Consequently, predicted thresholds in the visual condition rise precipitously as opposite cells are given more weight (*Figure 6B*, red curve). From these data, we can see that a Readout Index value of ~0.5 produces roughly matched visual and vestibular heading thresholds, similar to the animals' behavior (arrow in *Figure 6B*). With this weighting of opposite cells, the selective decoding model predicts similar heading thresholds for vestibular (2.33° ± 0.04°, mean ± SEM) and visual (2.40° ± 0.1°) conditions, as well as thresholds in the combined condition (1.58° ± 0.01°) that are very close to the optimal prediction (1.57° ± 0.02°, *Figure 6B*, green and brown curves). Critically, we note that the value of the readout index (0.5) that allows the selective-decoding model to mimic behavioral performance is the same value that we separately found to produce a pattern of CPs that approximates the experimentally observed data (*Figure 5C,G*). Thus, converging lines of evidence suggest a readout in which opposite cells contribute, but substantially less than congruent cells.

The above simulation results were based on an arbitrarily sized population of neurons (n = 1000). Not surprisingly, as the population size is varied in the simulations, predicted psychophysical thresholds decline as a function of population size in all stimulus conditions (*Figure 6C*). In our simulations, performance reaches a plateau at a population size of a few hundred neurons, and is roughly comparable to the animals' behavioral performance over a broad range of pool sizes. It must be noted that the extent to which performance asymptotes with population size is likely to depend on the exact structure of correlated noise, the extent to which the true decoder has full and accurate knowledge of the correlation structure, the extent to which correlated noise mimics stimulus variations and thus can be removed by decoding, and the degree of heterogeneity of the tuning curves in the population

(*Ecker et al., 2011*; *Beck et al., 2012*). Importantly, however, the key experimental features that we have sought to understand here are related to the sign of CP for opposite cells and not simply the magnitude of CPs; thus, the basic qualitative nature of our findings is not likely to be altered by the considerations above.

## Discussion

We explored the relative roles of correlated noise and selective decoding in generating the pattern of CPs observed across a population of multisensory neurons. While it is well accepted that correlated noise is necessary to observe CPs in large populations of cortical neurons (*Nienborg and Cumming, 2010*; *Cohen and Kohn, 2011*; *Nienborg et al., 2012*), a critical question is whether CPs can also reflect selective decoding of neural responses. Both the pure-correlation model and the selective decoding model could account for the peculiar finding that opposite cells show CPs that are systematically >0.5 in the vestibular condition and <0.5 in the visual condition (*Gu et al., 2008*; *Chen et al., 2013*). However, three main features of our findings favor the selective decoding model over the pure correlation model. First, the pure correlation model predicts a pattern of correlated noise that is inconsistent with experimental data from area MSTd, whereas correlated noise in the selective decoding model depends on congruency of tuning in a manner that is similar to that exhibited by pairs of MSTd neurons. Second, the pure correlation model predicts that CPs for opposite cells in the combined condition should be bimodally distributed, which is inconsistent with data from MSTd and with a broad range of predictions of the selective decoding model. Third, with modest weight given to opposite cells, the selective decoding model predicts a pattern of psychophysical sensitivity across stimulus conditions that closely matches the behavioral data from monkeys. Together our findings indicate that, although correlated noise is essential to observe CPs in large neural populations, selective decoding can play important roles in shaping patterns of CPs and behavioral performance.

### Origins of choice probability

Choice probability measures the trial-by-trial correlation between the activity of a single neuron and perceptual decisions. While the measurement itself is straightforward, the interpretation of CPs has been varied and somewhat controversial (*Nienborg and Cumming, 2010*; *Nienborg et al., 2012*). One possible interpretation of a significant CP is that variability in the response of a sensory neuron drives variability in perceptual decisions across trials—this is the so-called 'bottom-up' interpretation (*Parker and Newsome, 1998*). If this were true, then CPs would at least partially reflect the contribution of each neuron to the decision, and would be shaped by selective decoding of sensory signals. Along these lines, some studies have suggested that the pattern of CPs observed reflects the strategy by which sensory signals are decoded to perform specific tasks (*Uka and DeAngelis, 2004*; *Purushothaman and Bradley, 2005*; *Gu et al., 2007*, *2008*). Somewhat analogous conclusions were drawn in a previous study which showed that 'detect probabilities' depend on tuning preferences in a change-detection task, which may also be compatible with the notion of selective decoding (*Bosking and Maunsell, 2011*).

An alternative (but not mutually exclusive) possibility is that CPs are mainly driven by top-down feedback signals from parts of the brain involved in making decisions (*Nienborg and Cumming, 2009*, *2010*). If this is the case, then the pattern of CPs need not be directly related to the way that sensory signals are decoded to perform a task. Regardless of the relative roles of bottom-up and top-down signals in generating CPs, it is broadly recognized that CPs should not be observable in large neural populations unless noise is correlated among neurons (*Shadlen et al., 1996*; *Cohen and Newsome, 2009*; *Nienborg and Cumming, 2010*; *Cohen and Kohn, 2011*; *Nienborg et al., 2012*). Indeed, a recent study (*Liu et al., 2013*) provided the first experimental evidence that the difference in magnitude of CPs between two brain areas coincides with a difference in the structure of correlated noise between areas.

The controversy regarding whether CPs can reflect decoding strategy or just correlated noise was recently resolved by an important theoretical study (*Haefner et al., 2013*), which shows that CPs are determined by both factors. However, whether the pattern of CPs in a population reflects decoding strategy or not will depend on the specific details of the decoding weights, correlation structure, population size, etc. This theory shows that the decoding weights could be inferred from CPs if the full structure of correlated noise is known with sufficient accuracy and precision.

Using multisensory heading perception as a model system, we show that the pattern of CPs exhibited by neurons in area MSTd is more compatible with a model in which both selective decoding and

correlated noise contribute to the generation of CPs than a model in which CPs are determined solely from correlations. Multisensory representations may have advantages for studying CPs because of the presence of neurons that show opposite tuning for the two cues. If such neurons are decoded as providing evidence in favor of either their visual or vestibular heading preference, then the sign of their CP (whether it is > or <0.5) may reverse depending on the decoding strategy. We suspect that this feature of our model system has provided us with additional leverage to dissociate models that emphasize correlations vs selective decoding.

In our selective decoding model, we assume that responses of all MSTd neurons, both congruent and opposite, are decoded relative to their vestibular heading preference. This accounts for opposite cells having CPs <0.5 in the visual condition, as seen in the real data. Why might responses be decoded according to the vestibular heading preference? One possibility is that this allows the system to estimate heading in a manner that is robust to the presence of moving objects in a scene. Indeed, we have recently shown that a strategy of decoding both congruent and opposite cells according to their vestibular preferences can provide a near-optimal solution to the problem of marginalizing over object motion in order to extract heading in a robust manner (*Kim et al., 2014*). Moreover, adjusting the relative weighting of opposite to congruent cells can allow the population code to tradeoff robustness to object motion against increased sensitivity during cue integration (*Kim et al., 2014*). Thus, the selective decoding strategy that we employ here may provide a flexible way to decode self-motion signals efficiently under conditions in which moving objects may or may not distort optic flow.

## Materials and methods

### Tuning curves for hypothetical and real neurons

Hypothetical neurons

Two populations of MSTd-like neurons were simulated with cosine heading tuning:

$$Firing\_mean_{k,i}(\theta) = A \times (\cos((\theta + P_{k,i}) \times \pi/180) + 1), \tag{1}$$

where $k$ is the stimulus condition (visual, vestibular), $i$ indexes a particular neuron, and $\theta$ denotes heading direction within the horizontal plane [−180° +180°]. To simulate the heading discrimination task (*Gu et al., 2007, 2008*), we used a small range of headings [$\theta = \pm8°$, $\pm4°$, $\pm2°$, $\pm1°$, $\pm0.5°$, $\pm0.25°$, $\pm0.1°$ and 0°] around straight ahead. $P_{k,i}$ denotes the heading preference of each neuron, which is either +90° (rightward) or −90° (leftward) for the purposes of simulating heading discrimination, given that most neurons have monotonic tuning over this range (*Gu et al., 2008*). Each model population contains equal numbers of units with leftward and rightward heading preferences. $A$ is a scaling factor to adjust the peak response amplitude, and was arbitrarily set to be 100 such that the peak response of each neuron is 200 spikes/s and the minimum response is 0 spike/s.

Real neurons

Data were acquired from 129 single MSTd neurons that were recorded previously while animals performed a fine heading discrimination task based on either vestibular or visual (optic flow) cues, as well as the congruent combination of these cues (*Gu et al., 2008*). In each trial of the discrimination task, the monkey experienced forward motion with a small leftward or rightward component. At the end of each trial, the monkey made a saccadic eye movement to report his perceived motion as leftward or rightward relative to straight ahead. Across trials, heading was varied in fine steps around straight ahead. Nine logarithmically spaced heading angles were tested for each monkey, including an ambiguous straight-forward direction (monkey A: ±9°, ±3.5°, ±1.3°, ±0.5° and 0°; monkey C: ±16°, ±6.4°, ±2.5°, ±1°, and 0°). Because the tested headings were different between the two animals, these measured tuning curves were linearly interpolated to 0.1° resolution, and a set of new local tuning curves was constructed that had a common set of stimulus headings for the two animals [±8°, ±4°, ±2°, ±1°, ±0.5°, ±0.2°, ±0.1° and 0°]. This allowed data to be pooled across animals and decoded as a single population.

### Correlated noise

Correlation matrices that describe interneuronal noise correlations for a simulated population of neurons were constructed by assigning correlated noise ($r_{noise}$) to each pair of neurons (either hypothetical or real) according to the measured relationship between $r_{noise}$ and the signal correlation between the

pair of tuning curves, that is $r_{signal}$ (**Gu et al., 2011**). Signal correlation, $r_{signal}$, was computed as the Pearson correlation coefficient between the tuning curves (mean firing rates) for a pair of neurons. We previously showed that the relationship between $r_{noise}$ and $r_{signal}$ for MSTd neurons could be well described by **Gu et al. (2011)**:

$$r_{noise,i,j} = a_{vestibular} \times r_{signal,vestibular,i,j} + a_{visual} \times r_{signal,visual,i,j}, \qquad (2)$$

where $a_{vestibular}$ was 0.12 and $a_{visual}$ was 0.09 (**Gu et al., 2011**). For population decoding simulations based on real neurons (**Figures 4–6**), correlated noise between pairs of neurons was assigned according to **Equation 2** with the parameters above. For simulations based on hypothetical neurons, both coefficients in **Equation 2** were set to 0.1 for simplicity.

Single-trial responses of model neurons to each heading stimulus were generated according to the assumption of proportional Gaussian noise, with response variance set to be 1.5 times the mean firing rate to approximate the general behavior of cortical neurons (**Shadlen et al., 1996**; **Gu et al., 2008**; **Cohen and Newsome, 2009**). To generate simulated population responses that incorporated correlated noise among neurons, we incorporated the estimated correlation matrix and generated population activity for each stimulus modality ($k$), trial ($tr$), and heading ($\theta$) according to the following equation (**Shadlen et al., 1996**; **Cohen and Newsome, 2009**):

$$response_{k,tr}(\theta) = <response_{k,tr}(\theta)> + Q \times r_{rand} \times \sqrt{1.5 \times <response_{k,tr}(\theta)>}, \qquad (3)$$

where $Q$ is the square root of the correlation matrix, $r_{rand}$ is a random vector of standard normal deviates with the same length as the number of neurons (Matlab function 'normrnd', zero mean, unit variance), and '<>' represents the mean value. We typically produced 200 trials of responses for each heading ($\theta$).

## Analysis of visual-vestibular tuning congruency

As shown previously, neurons in area MSTd have vestibular heading tuning that can be either congruent with or opposite to their visual heading tuning (**Gu et al., 2006**, **2008**). Over the narrow range of headings used in the discrimination task, congruent cells generally have monotonic visual and vestibular tuning curves with matched slopes, whereas opposite cells generally have oppositely signed slopes (**Figure 2A**). To quantify tuning congruency, we compute a Pearson correlation coefficient between firing rate and heading for each tuning curve. From these, we compute a Congruency Index for each neuron, which is the product of the correlation coefficients for the visual and vestibular tuning curves. Thus, congruent cells will have a positive Congruency Index, whereas opposite cells with have a negative Congruency Index.

A pair of MSTd neurons can therefore have matched congruency (congruent–congruent pairs or opposite–opposite pairs), or mismatched congruency (congruent–opposite pairs). We categorized each pair of neurons as matched or mismatched by computing the product of their two congruency indices. Specifically, matched congruency pairs were classified as those having a product >0.2 (black symbols, **Figure 3A**), and mismatched pairs are those having a product <−0.2 (gray symbols, **Figure 3A**). The remaining cell pairs with products of Congruency Indices that fall in the range from −0.2 to +0.2 were classified as 'undefined' (open symbols, **Figure 3A**). These criteria are rather stringent, but we found that they reliably classify cells pairs.

To test whether a distribution of CPs contains a single mode (unimodal) or two modes (bimodal), we used a multimodality test based on the kernel density estimate method (**Gu et al., 2006**; **Takahashi et al., 2007**). Watson's U2 statistic, corrected for grouping, was computed as a goodness-of-fit test statistic to obtain a p value through a bootstrapping procedure. This test generates two p values, with the first one ($p_{uni}$) for the test of unimodality and the second one ($p_{bi}$) for the test of bimodality. If $p_{uni}$ >0.05, the distribution is defined as unimodal. If $p_{uni}$ <0.05, the hypothesis of unimodality is rejected. If $p_{bi}$ >0.05 as well, the distribution is considered bimodal.

## Population decoding

To transform responses from a population of neurons into quantitative predictions of behavioral sensitivity and choice probabilities, we decoded simulated population activity by computing a likelihood function as in previous studies (**Sanger, 1996**; **Dayan and Abbott, 2001**; **Jazayeri and Movshon, 2006**; **Gu et al., 2010**; **Fetsch et al., 2011**). For each stimulus modality ($k$), the likelihood over heading ($\theta$) given the observed population activity on a particular trial ($tr$) was given by:

$$Log\left(L_{k,tr}(\theta)\right) = \sum_{i=1}^{n} response_{k,tr,i}(\theta) \times Log\left(< response_{k,tr,i}(\theta) >\right)$$
$$- \sum_{i=1}^{n} < response_{k,tr,i}(\theta) >. \tag{4}$$

The first term describes the summation of each cell's contribution to the log likelihood function, which corresponds to the response on each trial weighted by the logarithm of each cell's tuning curve. The second term is the sum of all tuning curves, to counter biases associated with a non-uniform distribution of stimulus preferences.

This formulation embodies two assumptions. First, it assumes Poisson firing statistics. We have also tried alternative decoders that are based on Gaussian noise, such as the Fisher linear discriminant (*Dayan and Abbott, 2001*), and the results are almost identical. Thus, the details of the spiking statistics have little effect on our conclusions. Second, this formulation does not assume that the decoder has knowledge of the structure of correlated noise among neurons in the population, that is, a factorized decoder (*Averbeck et al., 2006*). While this assumption very likely affects the absolute sensitivity of the decoder, it is unlikely to alter the basic pattern of predicted results regarding CPs (*Gu et al., 2010*). However, further investigation is needed to systematically examine the detailed differences between decoders with and without knowledge of correlations, as well as the effects of different forms of correlated activity. Given these assumptions, our decoder is optimal in the maximum-likelihood sense, and does not otherwise assume a specific (and perhaps substantially suboptimal) 'pooling model', as was done in some previous simulations of CPs (*Shadlen et al., 1996*).

The decoder determined whether a tested heading was leftward or rightward relative to straight ahead by comparing the area under the computed likelihood function for leftward headings and rightward headings. If the summed likelihood for rightward headings was greater than that for leftward headings, the decoder would report 'right', and vice versa. Choice probability for each neuron in the simulated population was consequently computed for the ambiguous straight-forward heading (0°) (*Britten et al., 1996*; *Shadlen et al., 1996*; *Gu et al., 2007*, *2008*). The precision (threshold) of the decoder was also computed from each simulated psychometric function, which was analyzed using methods identical to those applied to the human and monkey behavior (*Gu et al., 2010*).

Heading information was decoded in two ways. According to our hypothesis that both congruent and opposite cells are decoded according to their vestibular heading preference (*Gu et al., 2008*), our main method of decoding involved using the vestibular heading tuning curve for each neuron in the formulation of *Equation 4*. In addition, for some analyses, heading was also decoded relative to the heading preference of each neuron in each stimulus condition. In this case, responses from the visual condition were decoded based on visual tuning curves, and so on.

For the selective decoding model, we implemented a function that controlled the contribution of each model neuron to the decoder output based on each neuron's congruency value, as given below:

$$weight_i = \frac{1 - e^{-(congruency_i/2) \times RI}}{1 - e^{-1}}. \tag{5}$$

Here, $weight_i$ denotes the decoding weight of the $i^{th}$ neuron, $congruency_i$ denotes the Congruency Index (described above) for the $i^{th}$ neuron, and $RI$ represents a Readout Index that ranges from 0 to 1 in steps of 0.1 (*Figure 4B*).

In the selective decoding model, the computed decoding weight of each neuron was then multiplied by that neuron's contribution to the likelihood function (*Equation 4*) before the likelihood contributions are summed across neurons. Thus, the computation of the likelihood under the selective decoding model was given by:

$$Log\left(L_{k,tr}(\theta)\right) = \sum_{i=1}^{n} weight_i \times response_{k,tr,i}(\theta) \times Log\left(< response_{k,tr,i}(\theta) >\right)$$
$$- \sum_{i=1}^{n} < response_{k,tr,i}(\theta) >. \tag{6}$$

# Additional information

### Competing interests

DEA: At the time of submission, DEA was a Reviewing editor for *eLife*. The other authors declare that no competing interests exist.

### Funding

| Funder | Grant reference number | Author |
|---|---|---|
| National Institutes of Health | EY017866 | Dora E Angelaki |
| National Institutes of Health | EY016178 | Gregory C DeAngelis |
| Recruitment Program of Global Youth Experts | | Yong Gu |
| Shanghai Pujiang Program | 13PJ1409400 | Yong Gu |

The funders had no role in study design, data collection and interpretation, or the decision to submit the work for publication.

### Author contributions

YG, Conception and design, Acquisition of data, Analysis and interpretation of data, Drafting or revising the article; DEA, GCDA, Conception and design, Analysis and interpretation of data, Drafting or revising the article

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
