## [Decision Letter]

Thank you for sending your work entitled “Contribution of correlated noise and selective decoding to choice probability measurements in extrastriate visual cortex” for consideration at *eLife*. Your article has been favorably evaluated by Eve Marder (Senior editor) and 3 reviewers, one of whom is a member of our Board of Reviewing Editors.

The Reviewing editor and the other reviewers discussed their comments before we reached this decision, and the Reviewing editor has assembled the following comments to help you prepare a revised submission. Your manuscript occasioned a significant discussion among the reviewers, largely around the issue of whether your paper makes a general contribution or whether it deals with a relatively narrow set of issues. Therefore, in preparing your revision, please think about any generalizations of the work that might make this more salient to people working on associated problems. Minimally, these issues should be confronted squarely in the revised Discussion.

General:

This paper focuses on the nature of choice probabilities (CPs). CPs is a measure of the covariation, in trial-by-trial-basis, between neural activity and behavior, typically in a decision task. As is well explained by the authors, the origins of CPs could originate from pure correlated noise of pairs of neurons as estimates of a neuronal population or by selective decoding, the neural response to a stimulus parameter(s) and its relation to behavior. According to the authors there is an historical debate of whether CPs, originate from pure correlated noise from neurons or by the selective decoding in a neuronal population. Here the authors use both simulations in artificial neurons biologically constrained and then tested the simulated results from real data of MSTd neurons, which are known by the authors; these neurons are associated with heading responses to visual flow and vestibular perturbations. The results of both simulations and real data are that MSTs neurons best conform to a selective decoding computation, although pure correlated noise contributes to selectivity.

Reviewer #1:

Found that the conclusions do not appear quite categorical since the authors claim that correlated noise is important but also cannot be excluded the selectivity decoding hypothesis. Curiously, cells in second somatosensory cortex have opposite tuning properties to the same stimulus, but that shared correlated noise has beneficial effect for improving the neurometric function and correlates with the psychometric performance (Romo et al., Neuron 2003). The beneficial effect came from a subtraction operation between the opposite tuning and correlated noise. Is this the case for congruent cells? In other words, for pool 2 neurons that carry the decision, she/he suspects that another group of neurons doing the opposite and very likely by subtracting noise and not a simple cancellation.

Reviewer #2:

As the authors acknowledge, the idea that all MSTd neurons are decoded according to their vestibular tuning preferences is a bit counterintuitive. He/she wonders whether the same simulations could be used to test the implications of this sort of decoding on the amount of information extracted about the stimulus. That is, does decoding only according to vestibular preference account for any sub-optimalties in the monkey's behavior? Such a result would provide more evidence for this decoding algorithm.

Reviewer #3:

This reviewer considers that the clear cut difference between the two models can only be formulated within the context of the Shadlen et al 96 model (recently revised by Haeffner et al), which the authors seem to take a ground truth, but which is itself a useful but fairly crude and simplistic picture of how the decision is being made), i.e. totally feedforward, no dynamics, etc).

Secondly even if one accepted that the decision making model is valid, the analysis does not allow the authors to draw general enough conclusions. Conceivably, one would set oneself to answer the question of whether it is possible to rule out the possibility than incongruent neurons do not contribute at all to the decision, regardless of all other modeling choices. The reviewer wasn't convinced by the analysis that this is the case. There are 3 main pieces of evidence: the patterns of CCs in Figure 3, the patterns of CPs in Figure 5 and the psychophysical data in Figure 6. In Figure 3, using an all-or-none approach where r_noise is ‘only’ a function of vestibular r signal, the data seems to favor the selective decoding model. However, (a) there are only 10 mixed pairs and (b) it is not at all clear that having r_noise be ‘mainly’ due to vestibular r_signal but with some contribution from visual r_signal, the two models would not look more similar to themselves and to data. This is a general concern: Even assuming the framework is valid, a simplified situation (in term of model parameters) is used to make categorical distinctions, and so it's not clear whether the conclusions are generic or if they only applied to the simplified situation.

The evidence from Figure 5 is also not shown to be conclusive/generic. The bi-modality in Figure 5 is not explained clearly and so it's not clear what exactly is necessary to obtain bi-modality. Is there no model with zero weights from incongruent neurons able to show unimodal CPs? Are there no parameters from the selective decoding model which would result in bi-modality? To make things worse, the selective decoding model seems to have one more degree of freedom (Readout Index). Similar concerns of generality apply to the analysis in Figure 6.

The authors conclude that both noise correlations and selective weighting probably contribute to the measured CPs. But this is hardly surprising (again, even assuming the whole framework is valid)!

In summary, the paper attempts to make a relatively minor distinction (i.e., to refute a model that represents a small region of parameter space (CPs are a result of only CCs) compared to the alternatives (CPs are a result of both CCs and weights)) and it does so it not in general terms, which limits the potential impact of the work.

---

## [Author Response]

Reviewer #1:

*Found that the conclusions do not appear quite categorical since the authors claim that correlated noise is important but also cannot be excluded the selectivity decoding hypothesis*.

We thank the reviewer for this comment as it is important to make these issues clear. Importantly, correlated noise is necessary to produce substantial choice probabilities in both models. So the key issue is whether the data suggest that there is selective decoding of responses of congruent and opposite cells. Another way to describe the key difference between the models is that the ‘pure correlation’ model relies on modality specificity of the noise correlation structure, whereas the ‘selective decoding’ model relies on modality specificity of the decoding weights. We have made a number of changes to the manuscript to clarify these issues, including additions and revisions.

*Curiously, cells in second somatosensory cortex have opposite tuning properties to the same stimulus, but that shared correlated noise has beneficial effect for improving the neurometric function and correlates with the psychometric performance (Romo et al., Neuron 2003). The beneficial effect came from a subtraction operation between the opposite tuning and correlated noise. Is this the case for congruent cells? In other words, for pool 2 neurons that carry the decision, she/he suspects that another group of neurons doing the opposite and very likely by subtracting noise and not a simple cancellation*.

In the case of [32], subtracting responses of oppositely-tuned neurons would be beneficial because these neurons generally showed positive noise correlations. In our studies, we consistently find that pairs of neurons with opposite tuning have negative noise correlations ([21]; [9]; Figure 3). This is the case for pairs of neurons with matched congruency (two congruent cells or two opposite cells), whereas mismatched pairs have weak noise correlation. Thus, it appears that the benefit of subtracting oppositely-tuned cells discovered by [32] would not be helpful in our case. We now mention this in the text where we introduce our correlation structure, and we cite the study of [32].

Reviewer #2:

*As the authors acknowledge, the idea that all MSTd neurons are decoded according to their vestibular tuning preferences is a bit counterintuitive*.

It is indeed counterintuitive when one thinks about the problem from the standpoint of cue integration. As we have shown previously (18; 16), decoding both congruent and opposite cells is suboptimal when the goal is simply to perform heading discrimination. However, we believe that other computational goals come into play, specifically the problem of dissociating self-motion from object motion. This is now discussed in more detail; decoding a mixed population of congruent and opposite cells may provide a near optimal solution to the problem of estimating heading in the presence of objects, which can be framed as a marginalization problem. In this light, such a decoding scheme becomes less surprising. We are currently working on another manuscript that describes these issues in much greater detail.

*He/she wonders whether the same simulations could be used to test the implications of this sort of decoding on the amount of information extracted about the stimulus. That is, does decoding only according to vestibular preference account for any sub-optimalties in the monkey's behavior? Such a result would provide more evidence for this decoding algorithm*.

We are unsure as to exactly what sort of sub-optimality the reviewer has in mind here, but we can say something about optimality of cue integration. As shown previously (16), incorporating opposite cells into the decoding impairs cue integration performance. On the other hand, including opposite cells adds robustness to object motion, and we have shown (in preliminary form) that titrating the proportion of opposite cells (i.e., the Readout Index) allows one to tradeoff cue integration against robustness to object motion (24). We now make this point in the Discussion and this will be explored in detail in a manuscript that is currently in preparation. Whether such a tradeoff in optimality of cue-integration would be observed behaviorally in the presence of moving objects is something that we cannot address at this time, but is something that we would like to study in the future.

Reviewer #3:

*This reviewer considers that the clear cut difference between the two models can only be formulated within the context of the Shadlen et al 96 model (recently revised by Haeffner et al), which the authors seem to take a ground truth, but which is itself a useful but fairly crude and simplistic picture of how the decision is being made) i.e. totally feedforward, no dynamics, etc)*.

It is true that our simulations do not specifically embody dynamics or feedback. However, our models are also not incompatible with such features. We simply compute a likelihood function over heading from the population responses of our model neurons. The correlated noise in our population responses might arise from either feedforward or feedback sources, and we are agnostic about that. Under the assumptions we make (see Methods), our decoder is optimal in the maximum likelihood sense. In this regard, we do not necessarily see our simulations as “formulated within the context of the Shadlen et al. 96 model”. That model adopts a specific pooling strategy for decoding responses, which is substantially suboptimal. We do not assume a specific pooling strategy but rather simply compute the likelihood over heading. We have modified the text to clarify this issue.

*Secondly even if one accepted that the decision making model is valid, the analysis does not allow the authors to draw general enough conclusions. Conceivably, one would set oneself to answer the question of whether it is possible to rule out the possibility than incongruent neurons do not contribute at all to the decision, regardless of all other modeling choices. The reviewer wasn't convinced by the analysis that this is the case. There are 3 main pieces of evidence: the patterns of CCs in*
Figure 3*, the patterns of CPs in*
Figure 5
*and the psychophysical data in*
Figure 6.

We agree that it is important to address the robustness of the conclusions. We have implemented new analyses to address this issue as detailed below.

*In*
Figure 3*, using an all-or-none approach where r_noise is ‘only’ a function of vestibular r signal, the data seems to favor the selective decoding model. However, (a) there are only 10 mixed pairs and (b)* it is not at all clear that having *r_noise be ‘mainly’ due to vestibular r_signal but with some contribution from visual r_signal, the two models would not look more similar to themselves and to data. This is a general concern: Even assuming the framework is valid, a simplified situation (in term of model parameters) is used to make categorical distinctions, and so it's not clear whether the conclusions are generic or if they only applied to the simplified situation*.

We have substantially improved this part of the analysis. In addition to just comparing the small samples of neurons that were classified as matched vs mismatched congruency, we now perform a model comparison using the data from all 127 pairs of neurons, which gives us substantially greater power. Specifically, we fit the data with two models: r_noise = a*r_signal_vestibular, and r_noise = a*r_signal_vestibular + b*r_signal_visual. Accounting for the extra free parameter, we find that the data highly favor the latter model which is consistent with the assumption of the selective decoding model (p=0.0003, sequential F-test). This section of the text has been extensively revamped, and the new results now appear in a modified Figure 3, while the original Figure 3 have become figure supplements.

*The evidence from*
Figure 5
*is also not shown to be conclusive/generic. The bi-modality in*
Figure 5
*is not explained clearly and so it's not clear what exactly is necessary to obtain bi-modality. Is there no model with zero weights from incongruent neurons able to show unimodal CPs? Are there no parameters from the selective decoding model which would result in bi-modality? To make things worse, the selective decoding model seems to have one more degree of freedom (Readout Index)*.

This is a fair point and we agree that it is important for the robustness of the different predictions to be addressed. We have now examined the bimodality of the predicted CP distributions for each model while we vary the key parameter of each model. The results show that these predictions are rather robust over a range of parameters of the two models that are consistent with the constraints implied by our other analyses as well. This is now addressed in a new paragraph as well as two new figure supplements (Figure 5—figure supplement 1 and Figure 5—figure supplement 2).

*Similar concerns of generality apply to the analysis in*
Figure 6.

The authors conclude that both noise correlations and selective weighting probably contribute to the measured CPs. But this is hardly surprising (again, even assuming the whole framework is valid)!

The parametric analysis of Figure 6, combined with the new parametric analyses of Figure 5—figure supplement 1 and Figure 5—figure supplement 2 suggests that only the selective decoding model accounts for all of the data reasonably well. Moreover, the range of Readout Index values that account well for the behavioral thresholds matches very well with the range (values around 0.5) that accounts well for the distributions of choice probabilities. Thus, we feel that the overall pattern of results provides considerable evidence that selective decoding makes a contribution to the choice probabilities that we observe experimentally.

*In summary, the paper attempts to make a relatively minor distinction (i.e., to refute a model that represents a small region of parameter space (CPs are a result of only CCs) compared to the alternatives (CPs are a result of both CCs and weights)) and it does so it not in general terms, which limits the potential impact of the work*.

We think that the additional analyses described above, as well as the clarified text in a number of places, have improved the manuscript substantially. Moreover, we do not agree that this represents a “minor distinction”. It has been argued in the literature that CPs may be determined solely by how neurons are correlated with each other, and that CPs may not at all reflect how neurons are decoded (we have colleagues who believe this very strongly). So the key issue at stake here, as we have now clarified in the manuscript (see other replies above), is whether there is evidence that the pattern of CPs in influenced by the manner in which neurons are decoded, and we think that the balance of evidence we present makes a reasonably strong argument in favor of this. As also discussed in the text, we think that our particular model system gives us additional power to test these predictions, due to the fact that opposite cells can be decoded either in favor of their visual or vestibular heading preference.